# Selective Subject Pooling Strategy to Improve Model Generalization for a Motor Imagery BCI [note 1]

**DOI:** 10.3390/s21165436

**Published:** 2021-08-12

**Authors:** Kyungho Won, Moonyoung Kwon, Minkyu Ahn, Sung Chan Jun

**Affiliations:** 1School of Electrical Engineering and Computer Science, Gwangju Institute of Science and Technology, Gwangju 61005, Korea; kyunghowon0712@gist.ac.kr; 2Korea Research Institute of Standards and Science, Safety Measurement Institute, Daejeon 34113, Korea; mykwon@kriss.re.kr; 3School of Computer Science and Electrical Engineering, Handong Global University, Pohang 37554, Korea; minkyuahn@handong.edu

**Keywords:** BCI, motor imagery, zero-training, selective training

## Abstract

Brain–computer interfaces (BCIs) facilitate communication for people who cannot move their own body. A BCI system requires a lengthy calibration phase to produce a reasonable classifier. To reduce the duration of the calibration phase, it is natural to attempt to create a subject-independent classifier with all subject datasets that are available; however, electroencephalogram (EEG) data have notable inter-subject variability. Thus, it is very challenging to achieve subject-independent BCI performance comparable to subject-specific BCI performance. In this study, we investigate the potential for achieving better subject-independent motor imagery BCI performance by conducting comparative performance tests with several selective subject pooling strategies (i.e., choosing subjects who yield reasonable performance selectively and using them for training) rather than using all subjects available. We observed that the selective subject pooling strategy worked reasonably well with public MI BCI datasets. Finally, based upon the findings, criteria to select subjects for subject-independent BCIs are proposed here.

## 1. Introduction

Brain–computer interfaces (BCIs) have shown great usefulness in facilitating communication for people with disabilities by extracting brain activities and decoding user intentions [1]. According to the control paradigm used, BCI systems can be categorized into three types: passive, reactive, and active [2,3]. Among them, motor imagery-based BCIs (MI BCI) are one of the most commonly used active BCI systems. MI BCI systems use the distinguishable characteristics in brain activity generated by imagining body movements, such as moving the left or right hand or foot. Thus, MI BCIs are more intuitive than other BCI systems that use reactive brain activity because they does not require external stimulation to generate the brain activity and can be applied easily in various areas to provide new communication and control channels for people who cannot move their own bodies [1]. Among other applications, these systems may be used to rehabilitate motor functions for patients [4] and develop game content [5,6]; however, some issues remain that must be resolved to increase the ability to use MI BCI. Generally, most BCI systems include two primary phases: calibration and testing. In the calibration phase, a certain number of data samples are collected and a classifier is trained or updated with the data [7]. In the test phase, users can actually operate the BCI application via their brain activity using the classifier that was trained or updated in the calibration phase; however, because the calibration phase is requires a long amount of time and is cumbersome in terms of actual use, researchers have attempted to reduce or skip the calibration procedure to improve user convenience. Several ideas and algorithms have been proposed to achieve reasonably comparable performance with relatively small training sample sizes (i.e., short calibration) or zero-training settings for various BCI systems [8,9,10,11,12,13,14,15,16]. Thus far, researchers have proposed a cross-session transfer model using the subject’s historic data [8,13] or a cross-subject transfer model using the data from other subjects [9,11,12,15,16,17]. Recent studies in the field of invasive BCIs have explored trial-to-trial variability within single-trial neural spiking activity by inferring the low-dimensional dynamics of neural activity [18,19]. In addition to within-dataset variability, one recent study addressed the cross-dataset variability problem by assigning training and test data from different datasets. Lichao et al. observed that cross-dataset variability weakened the learning model’s ability to be generalized across datasets, so they proposed a pre-alignment method to improve this [20] and ultimately achieved enhanced cross-dataset BCI performance.

In particular, for subject-independent BCIs (SI BCIs), robust machine learning algorithms and deep learning approaches have been proposed to extract common features across subjects so that new users can immediately operate BCI systems without the requirement of a calibration phase [16,21,22,23]. For example, to increase the model’s ability to be generalized across subjects, prototype common spatial pattern (CSP) filters were proposed based upon the observation that some CSP filters across different sessions or subjects are quite similar and could be clustered or combined as representative CSP filters from extracted filters [8,11,15], although notable differences between subjects may exist. In another study, an ensemble of classifiers that used subject-specific filters from other subjects was constructed, and the ensemble was sparsified with L1 regularization to enhance its ability to be generalized over subjects [9]. To improve the generalization ability of the model, most studies have explored feature (filter)-level selection methods rather than using all of the features (filters); however, in this work, subject-level selection methods are explored, as SI BCI performance is evaluated typically without appropriate selection at the subject level. In general, the SI BCI performance of each user is obtained by the following procedure: First, training data for the current subject are obtained by concatenating the training data from all subjects available, except for the current subject. Then, spatial filters or classifiers are trained on the concatenated training data. Finally, SI BCI performance is evaluated using the unseen test data from the current subject, and such cross-validation is referred to as leave-one-subject-out cross-validation (LOSOCV).

Training classifiers or spatial filters for all of the remaining subjects may be inappropriate if the subjects differ significantly or if some significant proportion of the subjects does not generate discriminative features during MI. It has been reported that approximately 10–30% of MI BCI users cannot achieve a reasonable classification accuracy necessary to operate a BCI system, referred to as BCI illiteracy [24,25]. In addition, those studies have reported that there are certain neurophysiological differences between subjects who perform well and poorly. Specifically, one study found a statistically significant positive correlation between MI BCI performance and alpha peaks at electrodes on the left motor cortex (C3) and the right motor cortex (C4), which can be expected as a potential alpha decrease during MI, showing that poor performers generated lower alpha peaks compared to good performers [24]. Another study found that a high MI BCI performance group (>70%) and low MI BCI performance group (<60%) showed a statistically significant difference between the theta and alpha band powers, indicating the significant correlation between MI BCI performance and such band powers [25].

In accordance with these previous studies, good and poor MI BCI performers have notably different characteristics, and thus training a classifier with selective features could enhance a classifier’s ability to be generalized to support session-/subject-independent BCI performance. In addition to selective features, selective subject use (only subjects who generate reasonable discriminative features) would improve SI BCI performance without developing sophisticated feature extraction algorithms. In this study, we propose a practical framework to select subjects to improve subject-independent BCI performance, rather than training with all subjects available. To the best of our knowledge, this work is the first to propose such a framework, which is described in Figure 1. In the framework, subject X′s MI BCI performance (f(X)) is evaluated using the filter and classifier (h(X)) trained with his/her own EEG data collected in the calibration phase, and then a decision is made whether the subject is a good source or not (g(X)). If g(X) returns true, the corresponding subject is added to the selective subject pool (S). Once the subject pool is created, new test subject X′s MI BCI performance can be evaluated using the trained filter and classifier (h(S)) with EEG data from the selective subject pool (h(S)) without the calibration phase. In this work, we suggest one possible decision method g(X) that can be applied in the proposed framework to determine whether the given subject is a good source or not and investigate its feasibility in evaluating cross-subject/dataset MI BCI performance.

This paper is organized as follows: In Section 2, the feature extraction methods are described briefly and our proposed selective subject pooling strategies that use subject-specific BCI performance are explained in a statistical sense. Further, the public MI BCI datasets used in this work are introduced in detail. Then, in Section 3, the comparative results of the selective subject pooling strategies are presented to investigate the feasibility of our proposed pooling strategy. Finally, certain issues raised in selective subject pooling are discussed.

We note that this work is an extended version of the conference article presented at the IEEE 9th International Winter Conference on Brain–Computer Interfaces (Gangwon, Korea, February 2021) [26]. In this previous article, we reported that a selective subject pooling method may enhance SI BCI performance in a public MI BCI dataset. In this work, we investigate the effect of selective subject pooling on SI BCI performance further using an additional MI BCI dataset and conduct an in-depth intensive investigation with further experiments.

## 2. Materials and Methods

### 2.1. Feature Extraction Methods

We used EEGLAB [27], the OpenBMI toolbox [28], and custom-built MATLAB codes to pre-process the EEG data, extract features, and evaluate classification performance. Among the various feature extraction methods, we used common spatial pattern (CSP) and multi-resolution filter bank CSP (MRFBCSP) methods in this work, which are believed to be basic and representative approaches to extract motor imagery features.

A common spatial pattern (CSP) is an optimized spatial filter that maximizes discriminative features from multi-channel data based upon recordings from binary class conditions [8,29,30]. A CSP is designed to maximize the EEG signal variance for one condition and minimize it for the another simultaneously. The variance in an EEG signal filtered by a band-pass represents its band power, and thus the CSP is an effective algorithm to identify ERD (event-related desynchronization) effects during motor imagery. It is understood well that motor activities such as actual or imagined hand movement attenuate the μ-rhythm [31,32] for several seconds, which is referred to as ERD. In applying a CSP to EEG data collected while imagining left and right hand motor movements, one may seek two groups of filters, where the first represents high variance (high band power) during left hand imagery, and low variance during right hand motor imagery, while the second filter group represents high variance during right hand imagery and low variance during left hand motor imagery. To apply the CSP algorithm, the segmented data were commonly band-pass filtered with cutoff frequencies of 8 and 30 Hz. From the trial-concatenated covariance matrix of dimension [C × T] (C and T are electrode channels and trial-concatenated time samples, respectively) for each class (left hand or right hand MI), the following generalized eigenvalue problem was derived to find the projection matrix *W* ∈ ℝ^C^
^× C^:(1)$$W^{T}\Sigma_{1}W=D\;and\;W^{T}\Sigma_{2}W=I-D$$
where Σ*_i_* represents a trial-concatenated covariance matrix of left or right hand motor imagery movement (referred to as class 1 and 2, respectively), *I* represents an identity matrix, and *D* denotes a diagonal matrix with elements in the range of [0, 1]. In the estimated projection matrix *W*, *W*′s first column vector represents the first spatial filter that has a relative variance of d_1_ (1st diagonal element of *D*), which maximizes the variance (band power) in trials for class 1. On the other hand, the last spatial filter maximizes the variance (band power) in trials for class 2. In practice, the first two or three CSP filters are selected for class 1 and the last two or three for class 2. For more detailed information on the CSP algorithm in a BCI system, refer to [8,29,30].

The multi-resolution filter bank common spatial pattern (MRFBCSP) is an extended version of FBCSP that increases the number of filter banks over three sub-decompositions over 8–30 Hz with four standard frequency rhythms (theta, mu, beta, and gamma) and five bands of a 6 Hz bandwidth (7–13 Hz, 13–19 Hz, …, 31–37 Hz) [10]. Compared to the conventional FBCSP, the MRFBCSP yielded better subject-independent BCI performance while maintaining a subject-dependent performance comparable to conventional FBCSP. In this work, the overlapping sub-decompositions of five bands of a 6 Hz bandwidth (10–16 Hz, 16–22 Hz, …, 34–40 Hz) were added to increase the amount of information within the sub-decomposition. From a total of 15 filter banks, (8–30 Hz, 4–7 Hz, 8–13 Hz, 13–30 Hz, 30–40 Hz, 7–13 Hz, 13–19 Hz, 19–25 Hz, 25–31 Hz, 31–37 Hz, 10–16 Hz, 16–22 Hz, 22–28 Hz, 28–34 Hz, and 34–40 Hz), 2 feature pairs were selected for each filter bank as usual, and, finally, the best feature pairs were chosen to create the MRFBCSP filter by the mutual information best individual feature selection (MIBIFS) algorithm, as in the conventional FBCSP [33].

### 2.2. Selective Subject Pooling

It is common to train subject-independent spatial filters and classifiers using all subjects available; however, in this work, we propose a selective subject pooling strategy that selects subjects who are likely to generate discriminative EEG patterns during motor imagery, and subject-independent classifiers are trained with the selective subject pool only. Subjects are chosen selectively according to subject-specific (SS) BCI performance, in that subjects are added to the selective subject pool when their SS BCI performance is better than a given performance threshold. To determine the threshold, a random statistical probability for binary classification [34] was introduced. According to [34], the statistical random probability in a binary classification problem is not 50%, but, more precisely, is 50% with a confidence interval at a certain *α* level (statistical significance) depending upon the number of trials (observations), which indicates that subjects A and B with different numbers of trials would have different statistical random probabilities, in which the random probability of trials classified correctly out of *n* trials and significance *α* is expressed theoretically as in [34]:(2)$$\tilde{p}\pm \sqrt{\frac{\tilde{p}\left(1-\tilde{p}\right)}{n+4}}z_{1-\frac{\alpha }{2}}$$
where $\tilde{p}$ represents the expected chance level and *z*_1−*α*/2_ represents the 1 − *α*/2 quantile of the standard normal distribution *N*(0,1). Applying this to left/right hand motor imagery BCI consisting of 100 test trials (50 per class), where the expected chance level, $\tilde{p}$, would be 0.5 and the theoretical lower and upper 0.95 (1 − 0.05) confidence limits would be 0.4039 and 0.5961, respectively. The upper limits of the theoretical random probability over various trials and several confidence levels are computed and illustrated in Figure 2.

For example, if one user achieved an accuracy of 0.6 in 100 trials, this is below the theoretical confidence limit for *p* < 0.01; however, it is above the confidence limit for *p* < 0.05. Thus, depending upon the confidence level and the number of trials, the reported accuracy could be considered to be at the random chance level or a level significantly higher than chance. It can be expected that such a limit can be used to determine whether the given subject is a good source or not. In this work, the upper confidence limit was used as a threshold (depending upon the number of trials and significance level) to determine whether the classification performance was appropriate to create a selective subject pool, and the feasibility of the selective subject training with respect to SI BCI performance was investigated. 

### 2.3. Experiments

Subject-specific evaluation, subject-independent evaluation with all subjects available, and subject-independent evaluation with the selective subject pooling strategy were compared to evaluate the selective subject pooling strategy’s efficacy. Two publicly available MI BCI datasets were used: Cho2017 [35] and Lee2019 [28]. These datasets have quite a large number of subjects (*n* = 52 and *n* = 54, respectively) compared to other open datasets and have been recorded with the use of sufficient electrode channels (*n* = 64 and *n* = 62 channels, respectively). Thus, we may observe broad performance distributions and reduce the probable bias that is attributable to small sample sizes.

For Cho2017 dataset, a total of 52 subjects (19 females, age 24.8 ± 3.86) performed a motor imagery experiment [35]. The Institutional Review Board of Gwangju Institute of Science and Technology approved the experiment (20130527-HR-02) and all subjects gave informed written consent before the experiment. EEG signals were recorded with a sampling rate of 512 Hz and were collected using 64 Ag/AgCL electrodes with the Biosemi ActiveTwo system. For each block, the first 2 s of each trial began with a black fixation and the indicative text (“Left Hand” or “Right Hand”) appeared for 3 s. Thereafter, the screen remained blank for 2 s and the inter-trial interval was set randomly between 0.1 and 0.8 s. The subjects performed motor imagery with finger movements with the appropriate hand when the indicative text appeared on the screen. The experiment consisted of five or six runs, each of which consisted of 20 trials per class (three subjects, s07, s09, and s46, performed only six runs). Visual feedback was not provided during each block, but the operator informed the subjects of their classification accuracy after each run. As a result, a total of 100 or 120 trials per class was collected. In this study, we divided the merged data into training (50%) and test (50%) sets. To evaluate offline performance, 21 electrode channels around the motor cortex (FC5, FC3, FC1, FCz, FC2, FC4, FC6, C5, C3, C1, Cz, C2, C4, C6, CP5, CP3, CP1, CPz, CP2, CP4, and CP6) were selected and downsampled to 128 Hz. The EEG data were then segmented into 1000 to 3500 ms time windows from the stimulus onset. For more detailed information about this dataset, see [35]. We note that the EEG data were segmented into 1000 to 3500 ms from the stimulus onset, although the indicative text appeared for 3 s and remained blank for 2 s. It was observed that the classification accuracy with 1000 to 3500 ms windows was higher than the accuracy with 500 to 2500 ms windows. Thus, it was expected that imagination of finger movement may continue for a short time even after the text disappeared.

For Lee2019 dataset, a total of 54 subjects (25 females, age 24.2 ± 3.05) performed binary class motor imagery (MI), event-related potential (ERP) speller, and four target frequencies steady state visual evoked potential (SSVEP) experiments in two different days [28]. The Korea University Institutional Review Board approved the experiment (1040548-KUIRB-16-159-A-2) and all subjects provided informed written consent before the experiment. In this work, we investigated motor imagery data collected on two different days. EEG signals were recorded with a sampling rate of 1000 Hz and collected using 62 Ag/AgCL electrodes with the Brain Products Brainamp system. For each block, the first 3 s of each trial began with a black fixation and an indicative arrow appeared for 4 s. Thereafter, the screen remained blank for 6 s (±1.5 s). The subjects performed motor imagery of grasping with the appropriate hand when the indicative arrow appeared on the screen. During the online test run, the fixation cross appeared on the screen and moved to left or right according to the EEG signal’s predicted output. During each day, the experiment consisted of two runs (training and test), each of which consisted of 100 trials per class. In this work, we merged the motor imagery data from the two different days for the analysis, which yielded the merged motor imagery data of 200 training trials and 200 test trials. To evaluate offline performance, 20 electrode channels around the motor cortex (FC5, FC3, FC1, FC2, FC4, FC6, C5, C3, C1, Cz, C2, C4, C6, CP5, CP3, CP1, CPz, CP2, CP4, and CP6) were selected and downsampled to 100 Hz. The EEG data were then segmented into 1000 to 3500 ms time windows from the stimulus onset. For more detailed information on this dataset, see [28].

**Subject-specific BCI evaluation.** The training and test data were divided for each subject. The training data were used to derive the spatial filters and linear classifier, and, finally, BCI classification performance was evaluated using the test data. This conventional approach is referred to as “subject-specific BCI (SS BCI) performance evaluation”. For the CSP algorithm, the offline EEG data were band-pass filtered between 8–30 Hz and segmented into time windows of 1000 ms to 3500 ms after the stimulus onset, which yielded [320 (time samples) × 21 (electrodes) × 100 (trials)] for training and testing from the Cho2017 dataset and [250 (time samples) × 20 (electrodes) × 200 (trials)] for training and testing from the Lee2019 dataset. After obtaining the CSP filters from the training data, the first and last two filters (a total of 4 filters) were selected, and classification was performed with the test data by regularized linear discriminant analysis (LDA) with automatic shrinkage selection [29,36]. For the MRFBCSP algorithm, feature vectors for each filter bank were estimated in the same way as the CSP and the best feature pairs were selected using the mutual information best individual feature selection (MIBIFS) algorithm. As addressed in [10], the MRFBCSP may extract more features than CSP, as the MRFBCSP extracts multiple CSP filters from multiple filter banks filtered with various bands. To evaluate performance, the best 6, 10, and 20 feature pairs were selected for comparison.

**Subject-independent BCI evaluation using all subjects available.** Unlike subject-specific BCI evaluation using individual subject data alone, BCI performance for each subject was estimated by leave-one-subject-out cross-validation (LOSOCV) with the available data for all subjects. Thus, one subject’s data were used for testing and all of the remaining subject data were used to train the spatial filters and linear classifier, and, finally, BCI classification performance was evaluated using the given subject data (test data). This approach is referred to as “subject-independent BCI (SI BCI) performance evaluation”. In this approach, we used a training data size of [320 (time samples) × 21 (electrodes) × 100 (trials) × 51 (other subjects)] for the Cho2017 dataset and [250 (time samples) × 20 (electrodes) × 200 (trials) × 53 (other subjects)] for the Lee2019 dataset. Then, CSP and MRFBCSP filters were extracted from these training data. For the CSP algorithm, the first and last three CSP filters (a total of 6 filters) were selected. For the MRFBCSP algorithm, feature vectors for each filter bank were estimated and the performance of the 6, 10, and 20 feature pairs selected were compared in a way similar to the subject-specific BCI evaluation approach.

**Subject-independent BCI evaluation using selective subjects.** We selected subjects and created a subject pool by introducing the selective subject pooling strategy using performance thresholds (upper limits of statistical random probability). Then, this subject pool was used to train the subject-independent spatial filters and classifier. This procedure (corresponding to g(X) in Figure 1) is described as follows: S1Determine the performance threshold according to the number of test data trials and statistical significance (e.g., *α* = 0.05, 0.01, …);S2Evaluate each subject’s SS BCI performance;S3Create a selective subject pool with subjects who achieve SS BCI performance (CSP-rLDA) greater than the performance threshold defined in S1. Note that depending upon statistical significance, the subject pool’s size varies (Table 1);S4Evaluate SI BCI performance using data from the selective subject pool. Note that when the current subject data are included in the selective subject pool, they are removed from the pool. Thus, LOSOCV is applied with the selective subject pool.

In addition, we investigated whether the proposed selective subject pooling strategy could be applied to datasets that have various numbers of trials. Because the threshold varies according to the number of trials as well as the significance level, we carried out a comparative analysis (including SS BCI and SI BCI performance) by changing the number of trials in the Cho2017 and Lee2019 datasets. As there were 100 training and test trials in the Cho2017 dataset, the number of sub-trials was set to 50 for that dataset. For the Lee2019 dataset with 200 trials, the sub-trial counts were set to 50, 100, and 150. To validate the results, sub-trials were selected randomly three times and the mean performance was used in the analysis. 

In this work, we compared SI BCI performance with various selective pooling strategies (various performance thresholds depending upon significance levels, *α* = 0.05, 0.01, 0.005, and 0.001, and various numbers of trials, *n* = 50, 100, 150, and 200). When we determined the performance threshold, the expected chance level, $\tilde{p}$, was set to 0.5 because the two datasets contained balanced and binary class motor imagery data, and the number of trials, *n*, was set according to the number of test trials. The performance thresholds used in this work and their corresponding sizes in the selective subject pool at various significance levels are tabulated in Table 1.

**Cross-dataset BCI evaluation using selective subjects.** We explored the cross-dataset BCI performance by training with one dataset and then testing with another to investigate the feasibility of using the selective subject pooling strategy to evaluate cross-dataset BCI performance. Thus, cross-dataset BCI performance for the Cho2017 dataset was evaluated by training the Lee2019 dataset and testing the Cho2017 dataset, and cross-dataset BCI performance for the Lee2019 dataset was evaluated by training the Cho2017 dataset and testing the Lee2019 dataset. For cross-dataset evaluation, the Cho2017 and Lee2019 data were preprocessed as follows: First, electrode channels were sorted in the same order across datasets, and then 20 electrodes near the motor cortex that all datasets had in common (FC5, FC3, FC1, FC2, FC4, FC6, C5, C3, C1, Cz, C2, C4, C6, CP5, CP3, CP1, CPz, CP2, CP4, and CP6) were selected. The remaining parameters, including the window length, frequency bands, and the number of training/testing data, were the same as in the case of SS BCI evaluation procedure.

## 3. Results

### 3.1. Subject-Specific and Subject-Independent BCI Performance

Subject-specific BCI (SS BCI) performance, subject-independent BCI (SI BCI) performance using all subjects available, and SI BCI performance using the selective subjects for the CSP and MRFBCSP in the given datasets, are listed in Table 2. Note that SI BCI-All refers to SI BCI performance using all subjects available, and SI BCI-*α* refers to SI BCI performance using various selective subject pools depending upon the significance level, *α*. The boldface text in Table 2 represents the highest SI performance among various *α* values for each feature extractor. With respect to MRFBCSP, only the cases with 10 feature pairs are displayed as there was no notable difference between 6, 10, and 20 feature pairs, and our main focus was not to find the best selection pair among filter banks, but to investigate the selective subject pooling strategy’s ability to enhance SI BCI performance.

In most cases, when SI BCI performance was compared to SS BCI performance, using training data from the same subject yielded better performance than using training data from the other subjects, which was as expected; however, in the case of 50 sub-trials in both the Cho2017 and Lee2019 datasets, all SI BCIs using MRFBCSP outperformed SS BCI slightly. 50 sub-trials are not expected to be sufficient to extract discriminative features among filter banks, while features extracted from different subjects may benefit from a sufficient number of trials despite the presence of inter-subject variability. Overall, MRFBCSP achieved better performance in subject-independent evaluation than did the CSP, which suggests that selecting a larger number of filter banks can help extract common discriminative features within as well as across subjects.

### 3.2. Selective Subject Pooling Strategy

In this study, we propose a selective subject pooling strategy that selects meaningful training subjects rather than training all subjects available and investigated its feasibility by comparing BCI performance over subject pools of varying sizes depending upon the threshold *α* (0.05 to 0.001) and the number of trials (50, 100, 150, and 200).

In Figure 3 and Figure 4, the performances of the SS BCI, SI BCI-All, and SI BCI-α methods are compared according to sub-trials and the extraction methods for the Cho2017 and Lee2019 datasets, respectively. Wilcoxon signed rank testing was performed to assess whether the difference between the SS BCI and SI BCI performances was statistically significant. If the null hypothesis was not rejected, the performance distributions of SS BCI and SI BCI do not differ significantly. In such cases, new users could skip the calibration phase because they are expected to achieve BCI performance as good as if the calibration phase is performed, although SI BCI performance may not be guaranteed to surpass SS BCI performance.

Consistently, we observed that SI BCI-α was better than SI BCI-All, as shown in Table 2 and Figure 3 and Figure 4. Clearly, a reasonable selection of good subjects may help improve SI BCI-α in performance compared to SI BCI-All; however, it was found that selective subject pools created at a higher significance level (subjects who performed far better) did not always achieve better performance. Performance improvement appeared marginal or varied slightly as the significance level increased, i.e., the number of selective subjects decreased gradually (see Table 2).

For Cho2017 dataset, as shown in Figure 3, with the CSP method, SI BCI-All yielded a performance distribution that differed significantly from that of SS BCI for both the 50 sub-trials and 100 trials settings (*p* = 0.0004 and *p* = 0.0002, respectively). On the other hand, the result for SI BCI-α=0.005 for 50 trials was comparable to SS BCI but differed significantly (*p* = 0.00002) from SI BCI-All. This indicates that SI BCI-α is applicable without the calibration phase and may achieve better performance than SI BCI-All (without selection) and comparable performance to SS BCI that requires the calibration phase, and thus, is promising. For 100 trials, we found that SI BCI-*α* = 0.005 demonstrated improved performance (from 0.5936 to 0.6102) compared to SI BCI-All, while the difference in the performance distributions between SI BCI-*α* = 0.005 and SS BCI was slightly significant (*p* = 0.0471). Looking at the difference in the performance distributions, SI BCI-All yielded 33 bad performers in BCI classification (lower than 0.6) and 9 good performers (higher than 0.7); however, SI BCI-*α* = 0.005 yielded 27 bad performers and 14 good performers. This observation implies that this selective subject pooling strategy may help improve SI BCI performance, and, in some cases, performance may be comparable to that in SS BCI. 

Compared to the CSP method, MRFBCSP appeared to extract common features across subjects successfully, and thus, the SI BCI-All performance distribution did not differ significantly from that of SS BCI for both the 50 sub-trials and 100 sub-trials settings. Thus, expectedly, there may be slight room to improve SI BCI-*α*; however, the SI BCI-*α* performances improved to 0.6442 (from 0.6155) and 0.6379 (from 0.6321) for the 50 sub-trials and 100 sub-trials settings, respectively. We note that one subject (s35) in the Cho2017 dataset showed ipsilateral CSP patterns so that the subject achieved high SS BCI performance, but the SI BCI performance was very poor, as the primary CSP patterns of most subjects were apparently contralateral during motor imagery. 

For Lee2019 dataset, as the Lee2019 dataset included 200 training and testing trials, the sub-trial settings varied between 50, 100, 150, and 200. As in Figure 4, with respect to the CSP method, there was considerable performance degradation in SI BCI-All, indicating that the distributions of SS BCI and SI BCI-All performances differed significantly at the 50, 100, and 200 sub-trials settings with *p* = 0.0399, *p* = 0.0307, and *p* = 0.0127, respectively. On the other hand, the performance distributions between SI BCI-*α* and SS BCI were quite similar for 50, 100, 150, and 200 sub-trials settings, while in the 100 sub-trials setting, SI BCI-*α* performed significantly better (*p* = 0.0001) than did SI BCI-All (Figure 4). With respect to the MRFBCSP method, the performance distributions between SI BCI-All and SS BCI did not differ significantly for all sub-trial settings, but SI BCI-All showed performance slightly inferior to SS BCI except for 50 sub-trials. Consistently, most selective subject pooling strategies increased SI BCI performance and yielded performance comparable to that in SS BCI.

Overall, across the two MI BCI datasets, we consistently observed that the selective subject pooling strategy improved SI BCI performance with various sub-trial settings and feature extraction methods. As reasonable subject pooling strategies, SI BCI-*α* and SS BCI may have comparable performance.

### 3.3. Comparison of CSP Filters

To investigate the effect of a selective subject pooling strategy at the feature level, we compared the CSP filters calculated from all subjects and subjects from the selective subject pools for each dataset. For simplicity, we present the first and last CSP filters only (Figure 5). As we selected electrode channels (21 or 20) around the motor cortex (see Section 2.1 to train the CSP, for illustration purposes, we note that the scalp topographic CSP filters were zero-padded for channels other than the channels selected (selected channels are presented as dots in the scalp topography). For Cho2017, when all subjects available were used to train the CSPs, we observed that the first (left hand MI) and the last CSP (right hand MI) did not show the contralateral activation pattern clearly, and thus the corresponding SI BCI-All yielded a performance that was quite inferior to SS BCI. On the other hand, the first and last CSPs for the selective subject pools created at *α* = 0.005 showed clear contralateral activations around electrodes on the left motor cortex (C3) and right motor cortex (C4), which are consistent with the neurophysiological findings [31,32]. In the case of Lee2019, we observed contralateral activations from the first CSP filters calculated from all subjects available. The CSP filters calculated from selective subjects at *α* = 0.001 showed more focal and contralateral activation around electrodes on the left motor cortex (C3) and right motor cortex (C4), but the difference was not as great as that observed with Cho2017.

### 3.4. Cross-Dataset BCI Performance

In addition to SI BCI performance, we investigated the selective subject pooling strategy’s feasibility to evaluate cross-dataset BCI. The cross-dataset BCI performance, including Cho2107 (train) to Lee2019 (test), and Lee2019 (train) to Cho2017 (test), is listed in Table 3 and illustrated in Figure 6. In terms of the Cho2017 cross-dataset BCI performance, training with Lee2019 did not benefit from the selective subject pooling strategy, as BCI performance with all subjects available in the Lee2019 dataset was 0.5545 and performance with selective subjects at *α* = 0.001 was 0.5623 for the CSP; however, for the MRFBCSP, the selective subject pooling strategy decreased the cross-dataset BCI performance slightly. Figure 6 shows that cross-dataset BCI performance in Cho2017 when CSP is trained with subjects from the Lee2019 dataset decreased BCI performance dramatically compared to SS BCI performance, and selective subject pooling did not improve performance. The MRFBCSP method showed similar results, although their BCI performance was better than that of those with the CSP method. On the other hand, for cross-dataset BCI performance of Lee2019, training with Cho2017 benefited highly from the selective subject pooling strategy. SI BCI performance using all subjects available in Cho2017 was 0.5988 and it increased to 0.6723 with selective subject training at *α* = 0.005. With respect to the MRFBCSP, it showed better ability to be generalized across datasets and better cross-dataset BCI performance compared to the CSP. The selective subject pooling strategy increased cross-dataset BCI performance from 0.6742 (all subjects available in Cho2017) to 0.6844 (*α* = 0.01), but the improvement of the MRFBCSP was not as great as with the CSP method (Figure 6).

These results may be inferred from the CSP filters shown in Figure 5. In the Cho2017 dataset, the CSP filters calculated from all subjects available showed no clear patterns, but the filters calculated from the selective subjects showed clear contralateral activation patterns. Hence, training all subjects available and only selective subjects may be quite different. On the other hand, in the Lee 2019 dataset, the CSP filters did not change as dramatically as in the Cho2017 dataset, and thus, training all subjects available and only selective subjects may be quite similar.

Different datasets have considerable variability attributable to various factors, including subjects, experimental environment, and device; thus, a selective subject pooling strategy or other approaches, such as pre-alignment methods [20], should be applied before evaluating BCI performance.

## 4. Discussion

The purpose of this study was to propose a selective subject pooling strategy to improve subject-independent MI BCI performance so that new users can use a BCI system immediately without the requirement of a lengthy calibration phase. To achieve this goal, BCI researchers have suggested approaches to extract robust features across subjects, such as approaches using Riemannian geometry [21] or deep learning [16,22,23]. At the same time, when using elegant feature extraction algorithms, a strategy that optimally maintains meaningful features alone and removes less important features has been proposed to allow the model to be generalized better across subjects [11,15]. In this context, we considered the selection strategy at the subject level.

With respect to the MI BCI system, several studies have reported that a significant number of subjects cannot achieve controllable performance [24,25] as the discriminative information from their brain signals is difficult to detect. In particular, a strong positive association between resting state alpha band power around the motor cortex and MI BCI performance has been reported [24], as well as significant differences between poor and good MI BCI performers in resting state alpha and theta band powers [25]; however, most MI BCI studies appear to train classifiers without selecting subjects.

Because feature extraction algorithms primarily use spectral band activities, extracting features from all subjects available without proper consideration may decrease the feature extraction algorithm’s ability to be generalized, so an appropriate subject selection method could improve the subject-independent BCI performance. One of the challenges in selecting subjects is heterogeneity in EEG data across sessions, subjects, and datasets (including different EEG devices and environments). This heterogeneity makes it difficult to obtain consistent criteria for subject selection, as the proposed criteria can be highly biased to a single dataset. In this study, we have proposed a selective subject pooling strategy that selects subjects based upon their SS BCI performance, selecting only those with SS BCI performance greater than statistical random probability. The method calculates CSP filters with these subjects alone. We found that selecting subjects using this selective subject pooling strategy extracted clearer discriminative patterns across subjects successfully and resulted in improved SI BCI performance compared to using all subjects available (without selective strategy). Moreover, these results were consistent in sub-trial settings (50, 100, 150, and 200 trials), which implies that our proposed selective criterion (thresholding) using statistical random probability depending upon the number of trials may be suitable for any dataset. In addition to SI BCI performance, the proposed strategy may sometimes be applicable to cross-dataset BCI, implying that selecting subjects appropriately may increase CSP’s ability to be generalized across datasets.

With respect to the threshold used for a selective subject pool, statistical confidence limits at the chance level (Equation (2) in this work) indicate the true statistical random probability of BCI performance [34]. In binary classification, the chance level can be expected to be 0.5; however, it is not precisely 0.5 in reality, as it depends upon the number of trials. With this reasoning, the chance level of 0.5 should be considered more carefully in combination with a confidence interval at the given significance level and the number of trials. It is understood that classification accuracy within the probability limit may not differ significantly from random in a statistical sense. This study has motivated us to use this random confidence limit as a threshold by increasing the significance level to select subjects who perform well. It is believed that this idea is rigorous and can be used generally to determine selective subject pools as it may provide a concrete standard across subjects and datasets that have various numbers of trials.

One can argue that the proposed strategy depends upon the extraction and classifier’s performance, which is one of its weak points. Compared to neurophysiological features, using classification accuracy as a threshold may be vulnerable because the results may vary depending upon the classifiers and extraction methods; however, as stated in Section 2.1, applying the CSP or its variant algorithm to band-pass filtered EEG signals is similar to finding contralateral ERD during MI, because the variance in band-pass filtered EEG signals is equal to the band power. Further, the features are trained using a linear classifier. Therefore, using SS BCI performance with a CSP and a linear classifier may be quite similar to scoring the contralateral ERD strength for each class. Figure 5 shows that a selective subject pool yielded CSP filter pairs that demonstrated clearer and more focal contralateral activation patterns compared to using all subjects available; however, we note that using a selective subject pool at a higher statistical significance did not always yield better SI BCI performance (Table 2). It may be expected from this that CSP filters calculated from higher thresholds (and thus fewer subjects) would be rather biased because of the use of limited subjects, and it would be difficult to generalize the features across subjects. As such, the features extracted may become more vulnerable to noise and inter-subject variability.

Moreover, we note here that the proposed methodology is not an ideal solution as there are several different approaches that can be applied to enhance subject-independent MI BCI performance, such as using different machine learning or deep learning models, different training methodologies, and different regularization or loss criteria; however, the simple framework provided here can be applied to improve a model’s ability to be generalized rapidly and easily. For example, researchers could develop new machine learning or deep learning approaches to enhance subject-independent MI BCI performance. Similarly, we wish to note that our proposed subject selection method may improve the performance of new machine learning and deep learning models. When one calculates an across-subject model for a new subject with data collected from other subjects already, one can extract a model from all past subjects. Further, one can extract a model from selective subjects who are likely to demonstrate meaningful features. In our proposed framework, we chose classification accuracy as the subject selection criterion; however, one can develop various other criteria based upon their classifier. Therefore, our proposed framework could be applied easily when researchers try to build a model from different datasets by scoring subjects who perform well to build their own classifier.

In this study, we applied our strategy to two public datasets, namely, the Cho2017 [35] and Lee2019 [28] datasets. As stated in Section 2.3 these datasets were selected because they recruited more than 50 subjects, covered all brain areas with electrode channels, and featured sufficient trial sizes, e.g., 100 and 200 training/test trials (including left and right hand MI). As our proposed strategy constructs selective subject pools from all subjects, a large number of subjects was required to investigate the selective subject pooling strategy’s feasibility because the proposed methodology’s underlying assumption is that behaviorally successful patterns of brain activity may be captured to build better subject-independent models, and patterns that arise similarly from most subjects should be weighted far more than those that arise exclusively from a few subjects. According to this reasoning, we have only found two public datasets that satisfy our requirements thus far. We note that selection from a small number of subjects may make it difficult to evaluate the generalization ability. To overcome this limitation without significant effort, reasonable data augmentation methods may be applied, and such investigations should be undertaken in subsequent works.

One possible concern in this study is subjects who poorly utilize BCIs. As we set the statistical random probability as the threshold for selective subjects, subjects who performed poorly (assuming lower than 0.6 because statistical random probability at *α* = 0.05 is 0.59) were excluded from training. Given that the CSP filters between good and poor performers differ, poor performers would not benefit from the selective subjects pooling strategy. To compare the effect of selective subject pooling on subjects selected and excluded (poor performers), we investigated SS BCI performance, SI BCI performance using all subjects available, and SI BCI performance using a selective subject pool for both the subjects selected and excluded (not shown here). For the subjects selected, the results showed the same trends as we observed in Figure 3 and Figure 4, where there was a large degradation in SI BCI performance using all subjects available compared to SS BCI performance, while using a selective subject pool increased SI BCI performance compared to using all subjects available. On the other hand, for the subjects excluded, there was no notable decrease in SI BCI performance when a selective subject pool was used, and, in fact, performance was even better than SS BCI performance in some cases, although it was around a random chance level. This result may imply that the features extracted from low/high performers in subject-independent MI BCI scarcely affect poor performers. In many other BCI studies, poor performers are not handled differently and most feature extraction algorithms have been applied in the same way for both poor performers and others, although they have been reported to have different frequency band activity [24,25]. To solve this issue, neuromodulation methods that change brain activity using an external stimulus, such as transcranial current stimulation [37] and neurofeedback training [38], may be especially effective if these methods minimize the differences in brain activities between poor and good performers.

While we primarily explored conventional CSP filters in this work, other variants of CSP filters [39,40] and various feature extractors [16,21,22] should be explored; however, our proposed selective subject pooling strategy is very simple and practical for application in existing cross-session, cross-subject, and cross-datasets, which may produce enhanced SI BCI classifiers. Moreover, it can motivate a new subject selection method (g(X)) that reduces the computational cost by removing redundant subjects at the beginning of the training or increases the generalization ability by augmenting good subjects in some way. Clearly, with respect to cross-session, cross-subject, and cross-dataset variability, further in-depth studies should be undertaken to enhance the generalization ability of the algorithm.

## Figures and Tables

**Figure 1 sensors-21-05436-f001:**
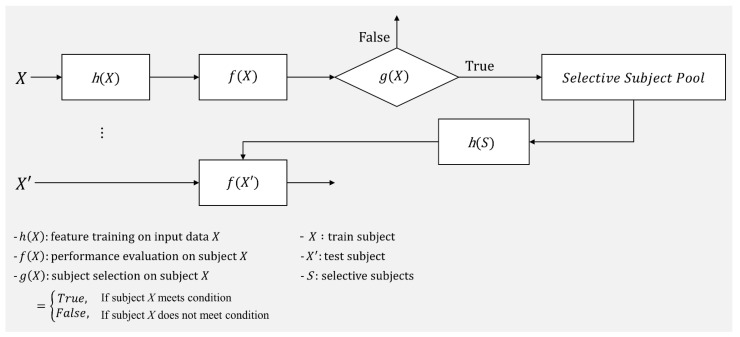
Selective subject pooling strategy for model generalization in motor imagery BCI. This represents the way the selective subject pooling strategy functions with h(X), f(X), and g(X). In this study, we introduce g(X) to increase the model’s ability to generalize for cross-subject/dataset evaluation f(X′) in motor imagery BCI.

**Figure 2 sensors-21-05436-f002:**
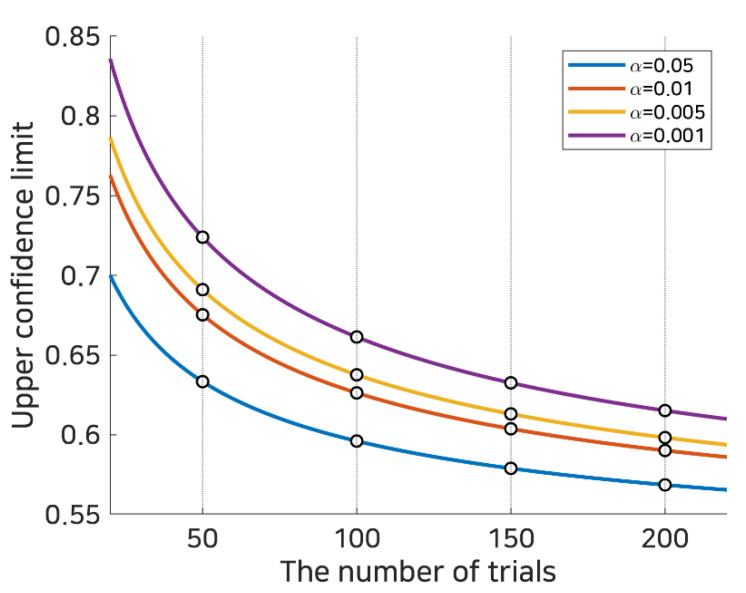
Random statistical probability. This represents the subject selection criteria using random statistical probability based upon the number of trials in binary classification. Each bold line denotes the statistical random probability calculated as in Equation (2) at statistical significance *α*, and the white dots on the dotted line indicate statistical random probabilities at 50, 100, 150, and 200 trials.

**Figure 3 sensors-21-05436-f003:**
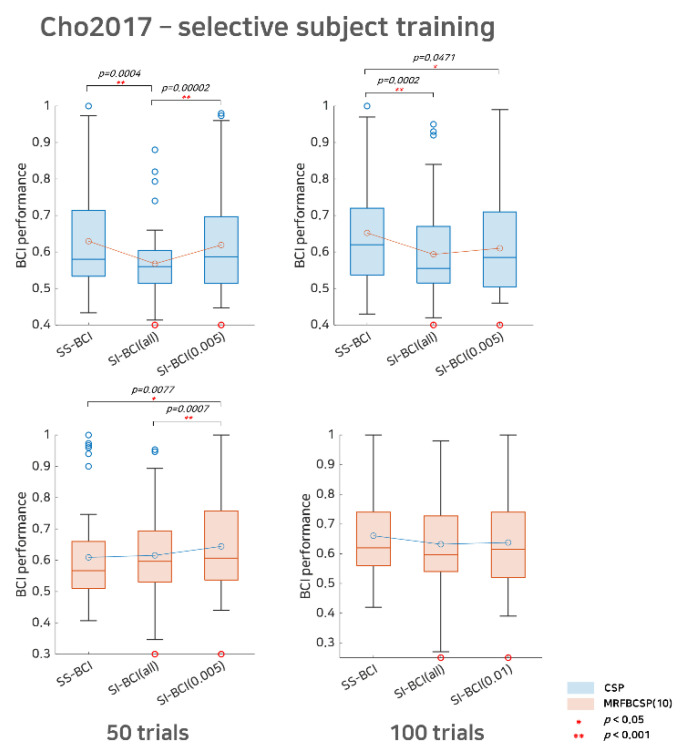
Comparative analysis of SS BCI and SI BCI performance with the Cho2017 dataset. This represents SS BCI performance, SI BCI performance using all subjects available (SI BCI-All), and SI BCI performance using selective subject training (SI BCI-*α*) for the Cho2017 dataset. Blue colored box plots indicate the CSP method and red colored box plots indicate the MRFBCSP method. Blue colored circles in the boxplot denote outliers and the red colored circle indicates a subject (s35) who showed ipsilateral activation for the CSP during motor imagery.

**Figure 4 sensors-21-05436-f004:**
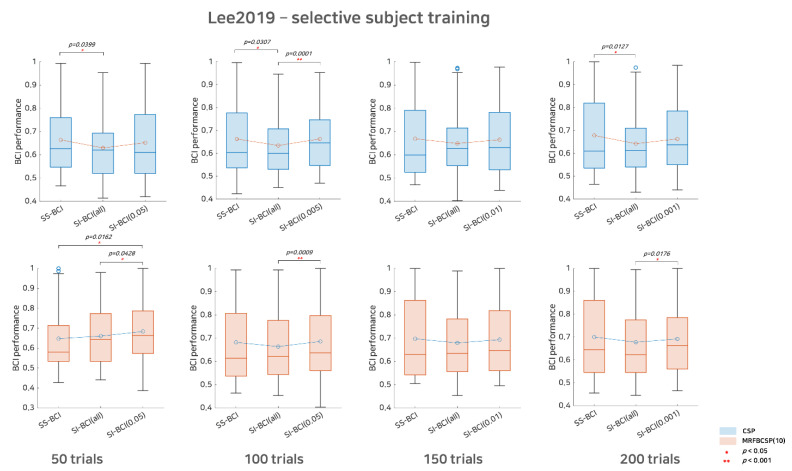
Comparative analysis of SS BCI and SI BCI performance for the Lee2019 dataset. This represents SS BCI performance, SI BCI performance using all subjects available (SI BCI-All), and SI BCI performance using selective subject training (SI BCI-*α*) for the Lee2019 dataset. Blue colored boxplots indicate the CSP method and red colored boxplots denote the MRFBCSP method. Blue colored circles in boxplots denote outliers.

**Figure 5 sensors-21-05436-f005:**
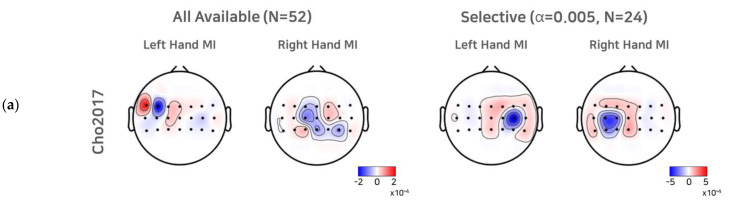
CSP filters created using all subjects available and selective subjects. For the Cho2017 (**a**) and Lee2019 (**b**) datasets, the topo plots represent the first and last CSP filters created using all subjects available and the selective subjects (*α* = 0.005, for Cho2017 and *α* = 0.001 for Lee2019).

**Figure 6 sensors-21-05436-f006:**
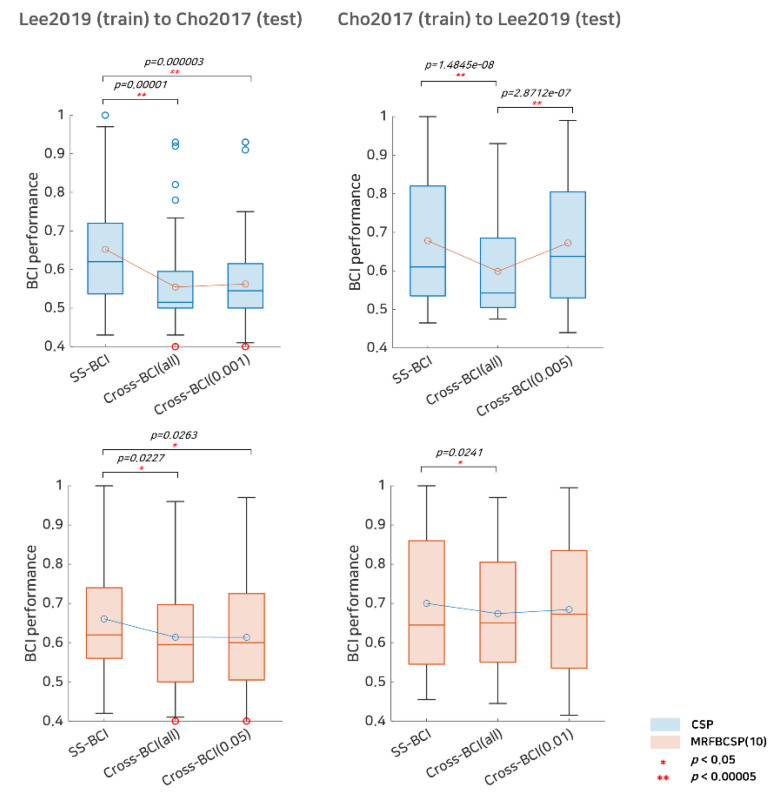
Comparative analysis of subject-specific BCI and cross-dataset BCI performance. This represents subject-specific BCI (SS-BCI) performance, cross-dataset BCI performance using all subjects available (Cross BCI-All), and cross-dataset BCI performance using selective subject training (Cross BCI-*α*) for each dataset. The blue colored boxplots indicate the CSP method and the red colored boxplots denote the MRFBCSP method. Blue colored circles in the boxplot denote outliers, and the red colored circles in Lee2019 (training) to Cho2017 (testing) indicates a subject (s35) who showed ipsilateral activation in CSP during motor imagery.

**Table 1 sensors-21-05436-t001:** Selective subject pool created using two MI BCI datasets.

*Dataset*	*Trials*	*α*	0.05	0.01	0.005	0.001
***Cho2017***	***50***	***Threshold***	0.633	0.675	0.691	0.724
		***#Subjects***	21	16	14	11
	***100***	***Threshold***	0.596	0.626	0.638	0.661
		***#Subjects***	31	26	24	20
***Lee2019***	***50***	***Threshold***	0.633	0.675	0.691	0.724
		***#Subjects***	24	18	16	14
	***100***	***Threshold***	0.596	0.626	0.638	0.661
		***#Subjects***	28	24	21	20
	***150***	***Threshold***	0.579	0.604	0.613	0.633
		***#Subjects***	33	27	26	24
	***200***	***Threshold***	0.5686	0.5902	0.5983	0.6152
		***#Subjects***	34	28	28	25

**Table 2 sensors-21-05436-t002:** Comparison of SS BCI and SI BCI performance.

*Dataset*	*Trials*	*Methods*	*SS BCI*	*SI BCI*				
				*SI-All*	*α = 0.05*	*α = 0.01*	*α = 0.005*	*α = 0.001*
**Cho2017**	**50**	**CSP**	0.6296	0.5678	0.6090	0.6087	**0.6191**	0.6182
		**MRFBCSP (10)**	0.6091	0.6155	0.6324	0.6319	**0.6442**	0.6340
	**100**	**CSP**	0.6522	0.5936	0.5904	0.6072	**0.6102**	0.6092
		**MRFBCSP (10)**	0.6607	0.6321	0.6316	**0.6379**	0.6369	0.6344
**Lee2019**	**50**	**CSP**	0.6642	0.6298	**0.6522**	0.6496	0.6460	0.6444
		**MRFBCSP (10)**	0.6477	0.6605	**0.6835**	0.6702	0.6753	0.6728
	**100**	**CSP**	0.6628	0.6343	0.6604	0.6580	**0.6630**	0.6629
		**MRFBCSP (10)**	0.6818	0.6638	**0.6859**	0.6821	0.6827	0.6823
	**150**	**CSP**	0.6692	0.6488	0.6605	**0.6651**	0.6644	0.6638
		**MRFBCSP (10)**	0.6975	0.6795	0.6898	**0.6936**	0.6933	0.6920
	**200**	**CSP**	0.6782	0.6421	0.6531	0.6488	0.6488	**0.6631**
		**MRFBCSP (10)**	0.7002	0.6774	0.6830	0.6894	0.6894	**0.6918**

**Table 3 sensors-21-05436-t003:** Cross-dataset BCI performance.

*Train*	*Test*	*Method*	*SS BCI*	*Cross-All*	*α = 0.05*	*α = 0.01*	*α = 0.005*	*α = 0.001*
***Lee2019***	***Cho2017***	**CSP**	0.6522	0.5545	0.5479	0.5547	0.5547	**0.5623**
		**MRFBCSP 10**	0.6607	**0.6138**	0.6136	0.6065	0.6065	0.6073
***Cho2017***	***Lee2019***	**CSP**	0.6782	0.5988	0.6454	0.6713	**0.6723**	0.6574
		**MRFBCSP 10**	0.7002	0.6742	0.6797	**0.6844**	0.6802	0.6794

## Data Availability

The data presented in this study are openly available in [GigaScience] at [10.1093/gigascience/giz002], reference number [28] and [GigaScience] at [10.1093/gi-gascience/gix034.], reference number [35].

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
