# Peer review of "Selective Subject Pooling Strategy to Improve Model Generalization for a Motor Imagery BCI [Author-notes fn1-sensors-21-05436]"

_sensors, 2021, doi:10.3390/s21165436_

Round 1
Reviewer 1 Report
- The paper focuses on a very important problem in BCI which needs to be discussed (even) more in the field: how to achieve top cross-subject and/or cross-session generalization? Typically this problem is tackled from the point of view of features selected or ML models used to learn how to combine those features in a general enough way. Robust and replicable features can be derived, or training methods can be used that force the ML model to combine such features (or even learn them entirely empirically) in a way that is robust instead of over-fit to the training domain. The authors take a different approach which is to focus on the subjects. It is well-known that brain activity indices used in BCI are variable across subjects. There are enough similarities between subjects to reveal some regularities in how different peoples' brains process the same task, but it is also true that different solutions in the brain activity space exist for successful completion of a task. Therein lies the problem - how does one build a model that can simultaneously pick up several distinct yet behaviourally-successful patterns of brain activity and recognise them equally well?
- The authors do a good job summarising previous efforts to achieve cross-subject and cross-session generalisation in BCI but in focusing on non-invasive/EEG approaches, miss out on important recent developments in latent space approaches which have been used in extracellular probe or ECoG papers. More concretely, I think at least the papers of Pandarinath et al 2018 and Degenhard et al 2020 are worth discussing because they're highly relevant to the issue at hand. The authors would be in their full right to disagree with me (and if so I would be interested in knowing why), but I personally believe the non-invasive space has a lot to learn and use from these two notable works.
- I believe the paper is well executed and explained. However my key problem with it is I do not see a sufficiently compelling reason why one would want to take this option over the more widely-adopted strategies of refining features, using different ML models and/or training methodologies and regularization/loss criteria. My main issue is that unless you can show to me that the subjects excluded from the pool had poor behavioural performance (did not understand or perform the task correctly), I think the proposed approach is a shortcut, in that it fails to provide a solution for subjects in which the BCI has poor accuracy. It is an interesting benefit to be able to, at least for a subset of subjects, provide a calibration-free BCI algorithm. But I would like to see a more compelling argument and rationale from the authors as to why one would go with this direction.
- I would also be interested in seeing a more granular description of the accuracy of the subject-independent-alpha approach at the individual subject level. Even the individual datapoints would suffice. I imagine in the test group (during the across-study cross-validation section) there are subjects in the test for whom the BCI works very poorly (they would resemble subjects which were excluded from the training pool) and others for whom the BCI works well. It would be useful to confirm this, which relates to my point 3 above.
Author Response
We appreciate the reviewer’s valuable comments. In the revised manuscript, we tried to address all comments. We believe that our manuscript is quite far improved. Point-by-point responses to each of the comments were listed below. We hope this revised manuscript is satisfying to reviewers.
Please refer to the attached file.

Reviewer 2 Report
In this manuscript a method for motor imagery BCI is proposed. The method is evaluated on 2 public available datasets on a subject- specific and subject independent scheme. This is a solid, comprehensive study. The manuscript is well-written with good structure. The objective is clear and scientifically sound and in this scope of the journal. I enjoyed reading it. Below are a few comments to the authors:
- The authors refer to their previous work presented in the IEEE 9th International Winter Conference on Brain-Computer Interface. In line 119 please revise “In this work” to “In the previously published work” to avoid any misunderstanding with the work presented in this manuscript.
- Please insert a flowchart of the proposed methodology.
- Since this work contain data from publicly available datasets, the authors should clearly state the novelty of the proposed study. The selective subject pooling strategy is first-time proposed?
Author Response

(The authors gave the same response as above.)

Reviewer 3 Report
This is a well-conducted research and a well-written paper. I only have minor comments:
1. In a number of cases you use acronyms that are defined after they are used (e.g. CSP line 61). Please fix these.
2. Along the same lines, in line 84 you use the electrode names C3 and C4 without identifying them as such, so it appears as jargon.
3. Figure 1 and lines 90-110 use the description of functions as g(X), h(X), etc. As you appropriately describe, these functions have very specific interpretations (e.g. h(X) is a classifier) -- I would suggest picking more appropriate names for these for readability, although it's up to you, this is a minor complaint -- I understood anyway.
4. Table 3 (train on one set, test on a different set) should also have the figure to go with it just as for Tables 1 and 2 for completeness.
Author Response

(The authors gave the same response as above.)
